# Family Farmers in Short and Long Marketing Channels: Lessons for Rural Development in Goiás, Brazil

Thiago de Carvalho Verano [1,*], Carlos de Melo e Silva Neto [2] and Gabriel da Silva Medina [3]

1   Agronomy School, Federal University of Goiás, Goiânia 74690-900, Brazil
2   Department of Research, Reference Center for Research and Innovation at the Federal Institute of Goiás, Goiânia 74594-111, Brazil; carlos.neto@ifg.edu.br
3   Faculty of Agronomy and Veterinary Medicine, University of Brasilia, Brasília 70910-900, Brazil; gabriel.medina@unb.br
*   Correspondence: veranoseco@gmail.com

**Abstract:** *Background*: Family farmers' access to markets is key for rural development. This study seeks to assess to what extent short and long marketing channels promote or inhibit the commercial inclusion of family farmers. *Methods*: The research was conducted in the Brazilian state of Goiás through questionnaires and interviews with rural outreach agents and family farmers' leaders. *Results*: The results reveal that 31.28% of sampled farmers are not included in any marketing channel. High inclusion rates in long channels (such as commodity markets) are related to high inclusion rates in short channels (such as farmers' markets), with some regions having greater availability of marketing channels than other regions. *Conclusions*: The high participation of family farmers in long channels linked to the cattle supply chain and agricultural commodities is related to the low participation of this category in other channels. Such results provide lessons for public policies by demonstrating the need to encourage a greater diversity of both short and long channels to greater marketing opportunities for family farmers.

**Keywords:** productive inclusion; coexistence of markets; distribution; short food supply chains; agri-food systems





## 1. Introduction

Family farmers' access to markets is deemed a relevant factor in tackling poverty [1]. The greater the access of rural families to domestic and global marketing channels, the greater their per capita consumption [2]. This effect is greater for households in short supply channels, but there is a complementary relationship between long and short channels in increasing the consumption of rural households [2]. Short channels (such as farmers' markets) are not the antithesis of long channels (such as commodities markets), as farmers move from one modality to the other or take part in both according to their profile [3].

Recent qualitative research identified aspects that promote or inhibit family farmers' participation in marketing channels. One aspect that relates positively to high farmer participation in short channels is the natural capital, in particular if they are close to large consumer centers [4]. Factors strongly related to improving family farmers' earnings in markets are the transportation infrastructure and access to market information [5].

Family farmers take part in global value chains in four manners: (1) buyer-driven supply chains—marked by the strong role played by retailers in determining quality and food safety standards; (2) producer-driven chains—in which family farmers receive less pressure regarding aspects related to food safety, since this responsibility is attributed to the processors; (3) bilateral oligopolies—where few and powerful leading companies establish contracts with farmers; and (4) traditional markets—with low entry barriers and where farmers do not need to meet many production standards [6].

Although the boundaries between short and long channels are unclear [7], they present structural differences. The short channels are characterized by direct sales from the producer to the consumer, while other links between the production and consumption stages mark the long channels. There is a prominence of business entities representing the different sectors in the long channels. The search for competitiveness is an ever-present goal in long channels, and the channel actors—suppliers, intermediaries, and producers—are in the surroundings of a company. In short channels, the empowerment lies with consumers, producers, and their relationships. In long channels, trust is established through contracts, while in short channels, trust is built through the interaction between the actors [8]. However, there are exceptions, such as the short channels linked to public procurement, in which transactions are performed by contract, and the long channels, such as sales to warehouses, in which contracts between farmers and intermediaries may not be used.

The agri-food model based on long channels arises from the demand for greater productive efficiency in agriculture and livestock production, and it changes as social groups start to direct their consumption towards sustainability. The new short channels arise as criticisms of the social and environmental impacts of long channels become more acute. Long and short channels present, to a certain extent, the transition towards sustainability as a recent guiding element of their emergence and transformations [9].

The agri-food model based on short channels does not work isolated and disconnected from the model based on long channels. Competition and convergence relations occur, making the short channels—depending on local configurations—sometimes antagonistic, sometimes alternative to the long channels [7]. These processes reconfigure the rural development patterns [7]. Therefore, short and long marketing channels make up the agri-food systems as family-based farmers and a wide variety of other farmers supply such systems.

The 2017 Brazilian Agricultural Census revealed that 71.83% of family farms in the Brazilian state of Goiás have the market as the main destination of their production, while 28.17% have consumption as the main destination of the goods they produce [10]. There is no precise information in the 2017 Agricultural Censuses on how the goods produced by family farmers are commercialized. Also, the scientific literature lacks quantitative studies that evaluate the economic relevance and the interaction between short and long channels.

This study fits this gap, as it assesses the possibilities of including family farmers in the short and long channels. It is intended to test the hypothesis that there are different forms of coexistence between short and long commercialization channels. Thus, it aims to understand to what extent the relationship between short and long marketing channels promotes or inhibits the inclusion of family farmers in markets. The specific objectives of this research are as follows: (1) to assess the occurrence of different short and long channels with the participation of family farmers; (2) to quantify the inclusion capacity of family farmers in these channels; (3) to spatialize the occurrence of the different channels in the assessed territory.

The academic literature on this subject reveals that the coexistence between the supply channels is neutral or inclusive [3,5,11,12]. However, there are places and contexts where this coexistence is conflicting and excluding [13–15]. Different agents and institutions of the agri-food systems are analyzed in these studies. In the present study, this topic will be evaluated specifically in consideration of family farmers' participation in the different marketing channels. This study aims to contribute to the literature with quantitative data, given that existing studies on this topic are essentially qualitative.

This article has the following sections in addition to this Introduction: (1) Literature Review, focused on the main studies that investigate markets and family farming; (2) Materials and Methods, dedicated to the description of the tools used and to the characterization of the geographical scope of the study; (3) Results, focused on presenting the data and findings of the investigation; (4) Discussion, to assess the novelty of the results based on the existing literature on the topic; and (5) Conclusions, composed of the main insights elabo-

rated from the discussion and the advances that this study presents concerning previous studies.

## 2. Literature Review

The literature on marketing channels is vast, diverse, and multidimensional. This section is structured as follows: (1) brief description of the concept of food orders and agri-food models; (2) the concept of short food supply channels; (3) differences between short and long channels; (4) types of coexistence between short and long channels; (5) critics of the concept of coexistence; and (6) data from the studied region (Goiás/Brazil) that contribute to the study of coexistence processes in that region. This study assessed the concept of coexistence based on empirical quantitative data on family farmers' participation in commercialization channels, surveyed in Goiás/Brazil and treated statistically.

Analyses of the production, distribution, and consumption processes of food, fiber, and other agri-food goods have resulted in major theoretical contributions. Agri-food orders and systems, global and local attributes, production of non-agricultural goods and landscapes, heterogeneities and rationalities, standardization and specificities, conventionalization and alterity, and hegemony, among other themes, are analytical lenses used to understand these processes [16]. Supply chains encompass production, distribution, and consumption stages, and the first critics of conventional agriculture focused on the production stage mainly. Over time, experts realized that sustainability should guide not only production but also distribution and consumption [17]. Thus, theorists, institutions, and social actors began to conceive two opposed agri-food models, the conventional and the alternative, each one with its specificities in the dimensions of production, distribution, and consumption [16,18–20] as summarized in Figure 1.

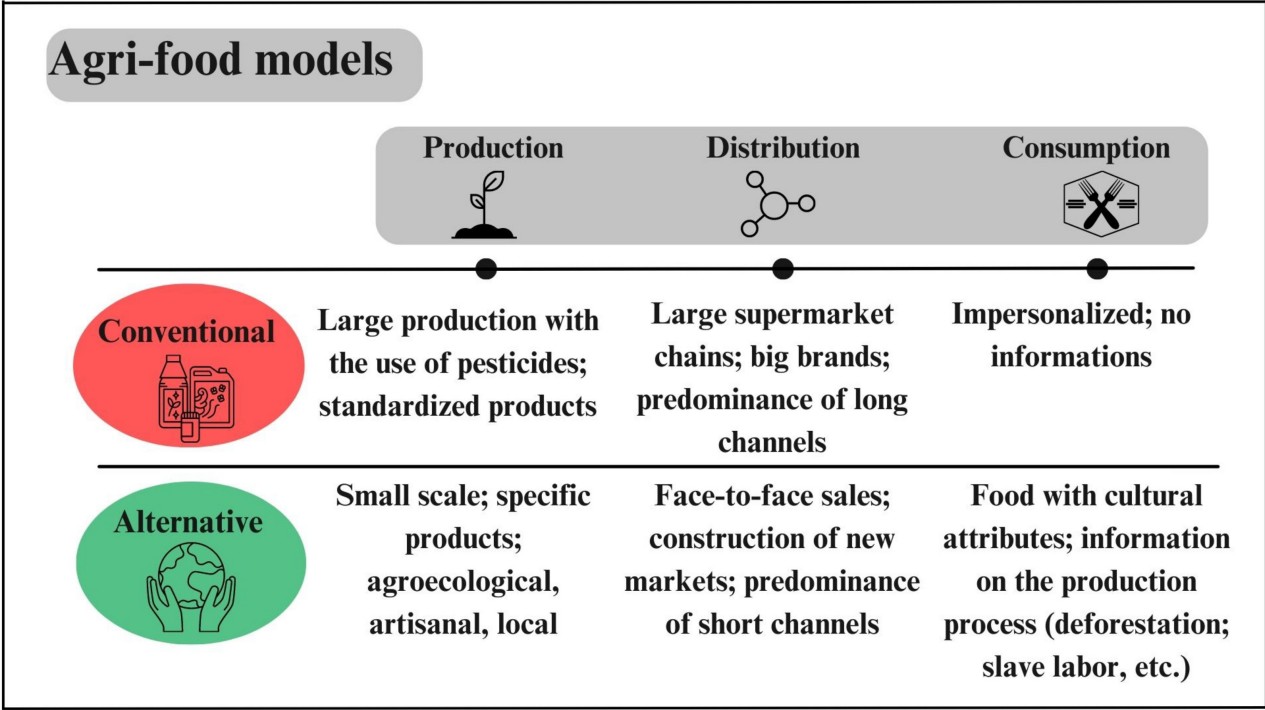

**Figure 1.** Differences between the agri-food models. Source: prepared by the authors based on Niederle; Wesz Junior, 2021 [16].

Alternative markets are a specific segment of global markets that emerge in the interest of conventional markets and interact with them [20]. Therefore, the alternative agri-food model does not stand in an opposed field to the conventional agri-food model [8]. The analytical dichotomy between conventional and alternative began to be overcome with

the development of agri-food systems [16]. The formation and development of an agri-food system are linked to the cultural aspects of farmers and consumers and the social and material conditions of the community or territory. The actors' capacity to organize themselves and to establish formal and informal rules and norms in an agri-food system is key to the effective participation of family farmers in markets and to the creation of added value and increased income for farmers [20]. In Brazil, legal and sanitary aspects, especially those related to the production of animal products, are a major obstacle for family farmers attempting to access more markets, as the legislation governing the production and marketing of these types of products was designed and is more adapted to regulating large-scale production [21].

Studies have questioned the sustainability of alternative food initiatives [22]. History has shown that experiments in organic production have been taken over by large economic groups in Europe and the United States [18]. This phenomenon has given rise to a new food order, which Niederle and Wesz Júnior (2021) call the aesthetic order, which has created, among other things, niche markets [16]. The circular bioeconomy is an approach to agri-food models that can contribute to solving the challenges of sustainability, as its approach combines issues such as biomass scarcity and waste management with issues such as food security, climate change, and the participation and protagonism of small producers [23].

Therefore, understanding the agri-food sector can help solve sustainability challenges. In an agri-food system, several social actors, institutions, norms, and sociocultural standards comprise elements from both conventional and alternative production, distribution, and consumption models. Figure 2 summarizes the diversity of agents, institutions, and organizations that comprise the agri-food systems.

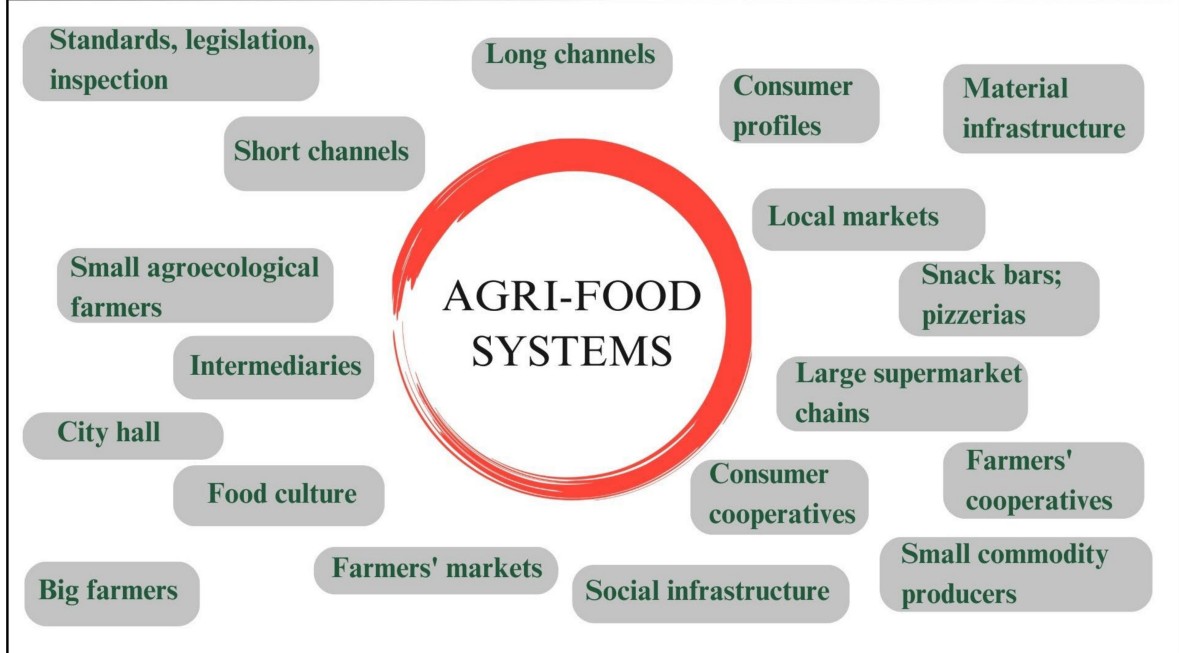

**Figure 2.** Agents, institutions, and organizations in agri-food systems. Source: prepared by the authors based on Niederle; Wesz Junior, 2021 [16] and Gazolla; Schneider, 2017 [24].

Therefore, some analytical lenses dichotomize, while others embrace convergences between production, distribution, and consumption types. Thus, the concept of food orders emerges to deal with the modalities of commercialization and aspects of production and consumption without adopting the dichotomy between short and long channels, not necessarily that of convergence and complementarity. The concept of food orders is a heuristic construct that reveals the heterogeneity of forms of production and consumption

and allows the identification of social dynamics that are difficult to analyze when using generalist explanations [16]. It considers the coexistence of several food orders—industrial, commercial, domestic, aesthetic, civic, and financial—which present "permeable" borders but are constantly disputed. This lens of analysis is not based on the dynamics of the production and distribution of a specific product but instead incorporates the logic of consumption more forcefully. In this way, neither production nor consumption is given priority in the analysis, as the focus is on the normative, regulatory, and cognitive mechanisms that shape social practices. [16]. To understand the differences between these two modalities, the theoretical framework with the dimensions of the short channels is presented in Table 1.

**Table 1.** Conceptual dimensions of short food supply chains and sources.

| Dimension | Theoretical Background |
|---|---|
| Short channels are a form of distribution that involves few or no intermediaries. | Deverre; Lamine, 2010 [25] |
| Short channels are those in which there is, at most, one intermediary between the producer and the consumer. | Chaffote; Chiffoleau, 2007 [26] |
| The difference between short and long channels is not simply the number of intermediaries or geographic distance but the supply chain's ability to generate connections between producer and consumer. | Marsden; Banks; Bristow, 2000 [27] |
| Short channels are described as capable of building value and meaning in the production–consumption relationship. | Ilbery; Maye, 2005 [28] |

The differences between short and long channels are subjected to greater scrutiny in the analysis. It is possible to differentiate them according to three dimensions: (1) objectives—short channels integrate and generate autonomy for actors, and long channels generate efficiency to the chain as a whole; (2) configuration—short channels bring producers and consumers closer together, and long channels have actors orbiting around the companies; (3) spatial relations—short channels produce products with territorial, local, or regional load, and long channels produce standardized products without regional differentiation [8].

Despite the aforementioned conceptual differences, many initiatives linked to the long channel model seek social and environmental attributes of products. Actors and organizations in the long channels have adopted environmental and social sustainability standards in their production and distribution processes, such as traceability, information on deforestation, child and enslaved person labor, and fair trade [8]. Both the short and the long channels have adopted the discourse of including family farmers in the markets to build quality attributes of their products.

The discussion on including or excluding family farmers in value chains received a significant contribution from Ros-Tonen et al. (2019) [29]. According to the authors, the economic, social, relational, and environmental dimensions underpin the concepts of inclusive business, value chains, and development. They claim that inclusive business and value chains practice the inclusion of some small farmers but with a main focus on the discourse of economic growth, while the concept of inclusive development is more anchored in promoting autonomy, agency, and conquest of rights of this social category. They conclude that to investigate the inclusion of family farmers in value chains, it is necessary to adopt the process with an analytical lens and that there are different ways to reach inclusion [29].

The participation of smallholders in contract farming, which is common in palm plantations in African countries, in the production of poultry and pork in Brazil, and the production of tobacco in some countries such as the Philippines, Brazil, and Mozambique, has been an object of study in recent years. Briones (2015) reviewed the literature that compares the gains obtained by smallholders involved in contract farming and those who

do not participate in this type of commercialization. This study used multivariate analysis (using tools such as linear regression and endogeneity correction) to conclude that contract farming in tobacco supply chains in the Philippines is increasing, that the gains of the farmer involved in this type of commercialization are higher, and that contract farming does not exclude smallholders [30].

The cohesion between the actors of a channel, which is markedly present in short channels, combined with the cohesion between segments, organizations, and institutions present in long channels, can contribute greatly to the food security of populations, even when subjected to natural disasters or compromises in the infrastructure of a region [11].

However, the coexistence of short and long channels does not guarantee the inclusive nature of the agri-food system, as they are not necessarily complementary. The inclusive character might be achieved through distinct marketing modalities composed of diverse actors and organizations, such as farmers, intermediaries, points of sale, supply centers, and consumers [12].

The search for interactions between actors and organizations of the long and short channels does not imply the loss of the characteristics of the short channels. The absorption of aspects of the long channels—quality standards, supply constancy, increased production efficiency, and increased number of customers—by the short channels should be accomplished in such a way that the initiative keeps its farmer–consumer connection. This reason leads scholars to conclude that the experiences of short channels have a maximum size to be reached, and it is more appropriate to replicate the initiatives than to expand them [31].

Recent studies on the coexistence of production, distribution, and consumption models are constructed under the lens of situations, types, and dimensions of coexistence. The situations of coexistence and confrontation are defined by (1) the actors and systems they are part of; (2) their interactions—rules, flows, etc.; (3) specific issues such as natural resources, quality criteria, and identity; and (4) the geographical or spatial clipping adopted [32].

Another tool used to investigate the coexistence of marketing channels is the typification of interactions. Non-cooperative coexistence is that in which short and long channels present low convergence of interests, low articulation between the actors of one and the other type of channel, and low dispute for market space. Competitive coexistence is based on a high degree of divergence between the channels, which compete for better market positions and seek to demonstrate attributes that add value to the product and attract more consumers. This type of coexistence references the dichotomous logic between short and long channels. Cooperative coexistence is the channels that present certain convergence of interests, articulation between actors and organizations, additional planning, or logistic operationalization. This type of coexistence occurs in places where the agents of the food systems act in both short and long channels, is justified in situations of the scarcity of different resources, and can promote food security in a given territory, and the need to add value to the channels is not high. In coordinative coexistence, there is a clear convergence of interests between channels and actors, and the search for value addition is a tonic of this interaction. The construction of sustainability attributes and social and cultural values occurs jointly between the short and long channels. In this coexistence, the interaction of processes and information between actors and organizations generates hybrid experiences that capture typical characteristics of the short and long channels [8].

The coexistence profile is based on three dimensions. The first refers to the tensions between specialization and diversification. Specialization, a unique characteristic of long channels, can also be seen in short channels when, for example, a farmer finds a niche market. On the other hand, diversification, which is usually attributed to short channels as a way to produce sustainably and that generates autonomy for family farmers, may reflect the lack of opportunities fundamental to the economic reproduction of the family. The second dimension is the innovation process. The productive innovations of the long channels have triggered major transformations in the agrarian spaces where agricultural modernization has taken hold. The social and territorial innovations of the short channels have given a voice to previously marginalized family farmers, allowing them the possibility

of coordination among actors and participation in decision-making bodies and marketing experiences. The dimension of adaptation concerns the capacities that actors of agri-food systems and their organizations have to maintain their production and social reproduction even in scenarios of uncertainty caused by instabilities in the economic, political, and climatic scenarios [9].

However, the concept of coexistence between different agri-food models can be deeply analyzed, especially in undeveloped countries. In a society where the main agents of the long channels dictate the legal phytosanitary standards, the menus and content of agrarian science courses, and food and agricultural policies, it is not possible to think of coexistence between different models, but rather co-presence [13].

The coexistence of agri-food models in Brazil presents some specificities not deeply discussed in the abovementioned studies. In our country, many family-based farmers—who were not legally categorized as farmers not long ago—do not have access to public policies. They have no or very low agricultural income, occupy small areas with impoverished soils, have access to little infrastructure, both productive and for the flow of production, and are involved in unfavorable sociopolitical arrangements [33]. In other words, coexistence between the small production-based model and the large production-based model—at least from a public policy perspective—did not occur since a large portion of the agents that were, in fact, part of the small production-based model—decapitalized and impoverished farmers—did not even receive the status of farmer.

Moreover, this coexistence was neither harmonious nor based on complementarity or the search for synergies. On the contrary, it has always been based on the favoring and economic, cultural, symbolic, and social exaltation of the actors, organizations, and structures linked to the long channel model. Although the Brazilian experience presents a relatively friendly coexistence between these models—with the creation of structures and policies aimed at the production and commercialization of family farming products between 2002 and 2016—what governs this coexistence is the dispute for territories, narratives, and attention from the State. In the political and institutional dimensions, the model based on strengthening family farming, agroecology, and alternative networks has always been opposed to the model of large-scale extensive production at first and intensive production today. This characteristic of the Brazilian experience raises the question of whether the mutually beneficial coexistence of these different agri-food models is possible over a long time in undeveloped countries where tensions arising from the advance of agrarian capitalism persist [14].

There is a distinction between the different production, distribution, and consumption models in Goiás. The state's south is the archetype of agricultural and agrarian modernity, while the north is seen as a poor region marked by the resistance of family-based farming communities. The south is home to the main commodity-producing regions, the highest level of agrochemical consumption, the highest productivity rates, and the best road and energy infrastructure. It is an area nationally recognized for the large-scale production of a small diversity of products. Most squatters, land reform settlers, and extractivists are concentrated in the northern portion. It is an area with higher rates of conserving Cerrado's natural resources but lacks road and energy infrastructure [34].

Causality relations were analyzed, and comparisons were made between channels with large and small participation of family farmers. Therefore, we chose to use the typification of coexistence proposed by Thomé et al. [7], but with some adaptations due to the scope of the research. Instead of using five categories of coexistence—cooperative, coordinative, competitive, and non-cooperative—we decided to typify the coexistence relations between the channels into (1) inclusive, (2) excluding, and (3) neutral.

## 3. Materials and Methods

This study was conducted in the Brazilian state of Goiás, which is one of the Brazilian references in extensive commodity farming and large-scale cattle ranching, which accounts for 11.3% [35] of the local GDP's Gross Added Value (GVA). But family farmers also play an

important role in the state of Goiás, accounting for 38% of the Gross Value of Agricultural and Livestock Production (GVA), according to IBGE (2017).

Using 13.81% of the area of the stage of Goiás, family farms represent 62.87% of the total rural households. Among Goiás rural households with agricultural production, 63.31% are family farms, which are responsible for 10.63% of the state's agricultural and livestock production value [36].

A partnership for data collection was established with the Goiás Agency for Technical Assistance, Rural Extension and Agricultural Research (EMATER/GO) since it has local offices in 201 of the 246 municipalities in Goiás. This institution provided technicians from the Local Units of the municipalities to locate and interview key informants capable of answering questions regarding the inclusion capacity of family farmers and income generation of each channel in this category. To this end, meetings were held with representatives of the state and regional coordination of EMATER to prepare the content of the interview script. Next, training meetings were held for the technicians to categorize the channels between short and long and to select the interviewees. In this way, each interview resulted from gathering information from the local technician and the different social actors representing the different channels. In Goiás, there are 201 municipalities with local EMATER offices. From this total of offices, we obtained interviews via Google Forms, with data regarding the occurrence and number of family farmers participating in each channel from 155 municipalities. Subsequently, telephone contact was established with ten local technicians to test the tool's accuracy. The contacts with the local technicians were important to attest to the interview's ability to translate the municipality's reality and build the second interview on gross income generation for farmers in each channel.

The second interview was sent to the 155 Local Units of EMATER that participated in the first stage of the research, but only 75 responded. Seventeen interviews were sent with incomplete answers, and 58 out of the 75 were used. To test the accuracy of the second interview and confirm some data, the researchers phone-called local technicians of the 58 municipalities that sent complete interviews. Therefore, both the first and second samples were non-probabilistic.

The first questionnaire was divided into two parts: long and short channels. The most frequent long and short channels in the state were listed, and the interviewees answered two questions for each channel: (1) whether or not family farmers are participating in the channel and (2) how many family farmers participate. In the second questionnaire—applied to the sub-sample of 58 municipalities—only one question was asked per marketing channel: how many Brazilian Reais do the group of family farmers in the municipality participating in the channel receive per month as gross income? From the data collected with this question, we obtained the sum of the gross productive income obtained by all the family farmers sampled in each channel. It was considered an income universe—100% of the income—the sums obtained by sales in all short and long channels. Therefore, this percentage variable was called productive income from each channel, which expresses the share of gross income obtained in each channel compared to the total income obtained in all channels. The data collected on income generation were inconclusive, as information was obtained only on the gross income obtained by farmers in each channel. In addition, it was impossible to assess the added value achieved by the farmers included in each channel or to quantify the farmers participating in more than one marketing channel. Figure 3 presents how data collection was conducted.

Thus, the survey obtained data regarding marketing channels and family farmers' participation in 63.01% of the municipalities of Goiás. The data were collected between August 2020 and March 2021. Respondents surveyed the data with information depicting the period before the COVID-19 pandemic. Therefore, the data presented in this study are for the period/year of 2019. Table 2 presents the types of short and long channels surveyed and their respective characterizations.

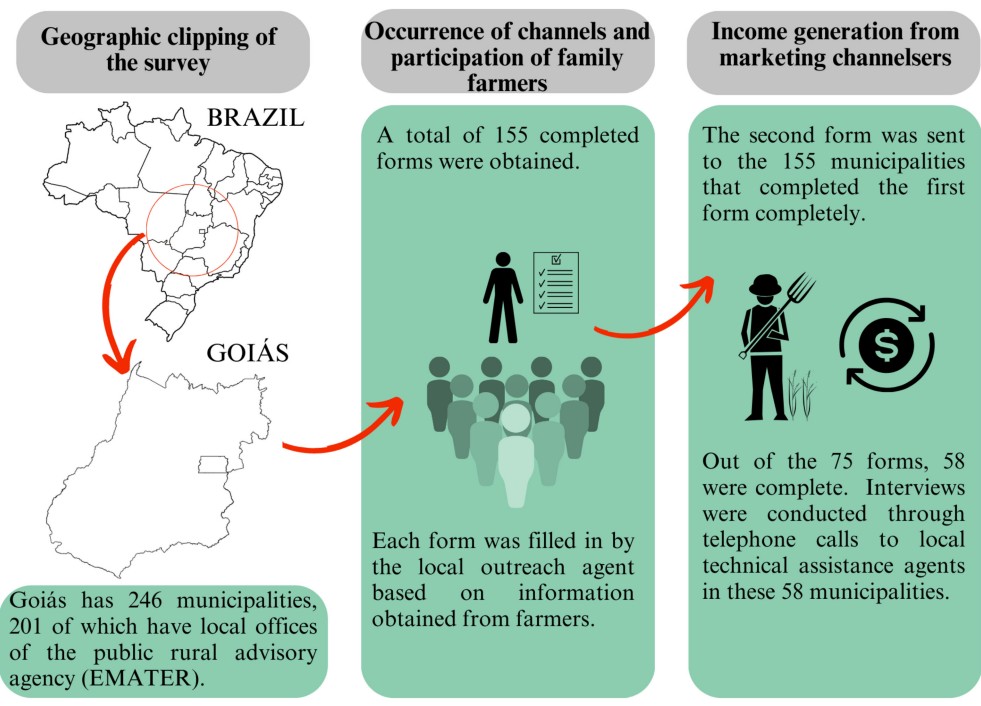

**Figure 3.** Geographic outline of the survey and data collection protocol.

**Table 2.** Types of short and long channels surveyed.

|  | Name | Abbreviation | Characteristics |
|---|---|---|---|
| Short channels | Neighbors | NEIGHBOR | Sale of products to community neighbors and adjacencies |
|  | Fair (farmers' market) | FAIR | Sale in a permanent physical space, managed by family farmer(s) where only family farming products are sold |
|  | Local butchers | BUTCHERS | Sale of live cattle to local butchers |
|  | Institutional market | PROCUREMENT | Institutional sale of products for school meals for public school students |
|  | Door to door | DOOR | Selling products door to door in the city or countryside |
|  | Sale on property | PROPR | Sale of products to the final consumer in the production unit itself |
|  | Apps and social networks | APLIC | Sale of products to the final consumer using virtual tools such as applications and social networks |
|  | Extractivism of Cerrado products | EXTRACTS | Direct sale to the final consumer of fruits, peels, and other products from the socio-biodiversity of the Cerrado |
|  | Stalls on city streets | STALL | Sale of products at stalls set up on the side of roads or on city streets |
|  | Innovation—new channels | INNOVATION | Commercial innovations; ways of marketing not described in the other channels; opening of new markets |
|  | FF specific store | STORE | Sale in a permanent physical space, managed by family farmer(s) where only family farming products are sold |
|  | Basket system | BASKET | Sale of product kits with defined periodicity between participating farmers and consumers |

**Table 2.** *Cont.*

| | Name | Abbreviation | Characteristics |
|---|---|---|---|
| | Milk to dairy | MILK | Sale of unprocessed raw milk to dairies or to carriers who resell to dairies |
| | Slaughterhouses | SLAUGHTERHOUSE | Sale of cattle to slaughterhouses or to intermediaries who resell to slaughterhouses |
| | Agricultural commodities | COMMOD | Sale of soy, corn, sorghum, or cotton that will be transacted via long channels |
| Long channels | Warehouses | WAREHOUSE | Sale of fruits and vegetables to supply centers or to intermediaries who resell at these centers |
| | Long-channel extractivism | EXTRATCL | Sale of fruits, leaves, bark, and other Cerrado socio-biodiversity products to middlemen |
| | Integration systems | INTEGR | Sale of poultry and pork via integration systems with large agroindustries |
| | Other long channels | OTHERCL | Other long marketing channels not mentioned above |

The fact that many farmers participate in more than one marketing channel made it impossible to collect information regarding how many family farmers participate in long channels and how many participate in short channels. Therefore, the variables quantity of short-channel and long-channel marketing outlets occupied by family farmers in each municipality were created; they were calculated by adding up the quantities of family farmers participating in each channel in the municipality. To arrive at the participation index of family farmers in each marketing channel, the universe in each municipality was considered to be the number of rural family households obtained from the 2017 Agricultural Census of the Brazilian Institute of Geography and Statistics (IBGE). To gauge the level of family farmers' participation in the municipality in short and long channels, a variable called the participation index of family farmers in the set of short and long channels studied was created. This variable was calculated through the ratio between the number of marketing outlets created by the set of channels and the number of rural family households in each sampled municipality. In some municipalities, the inclusion index was higher than 100% because family farmers participate in more than one channel and because more than one family resides in some rural family households. In these cases, the inclusion index was considered 100%. The sum of the participation indices of all short and long channels was calculated to calculate the percentage of family farmers who do not participate in any marketing channel. The difference between this value and 100 is the index of farmers not included in the marketing channels. The names and descriptions of the variables used are shown in Table 3.

**Table 3.** Variables used to estimate the percentage of family farmers participating in the studied markets.

| Name | Description |
|---|---|
| Marketing outlets | Number of family farmers participating in each channel, the set of short channels or the set of long channels |
| Participation index of family farmers | Percentage value obtained by dividing the number of family farmers participating in each channel, the set of short channels or the set of long channels, and the number of rural family households in the sampled municipalities |

These created variables reflect family farmers' participation in marketing channels at a given time (in the case of the present study, 2019). However, the inclusion and exclusion of this category are not a fact but a process. This study did not investigate such processes because, to meet the objective of developing a broad geographical area (the state of Goiás), it was necessary to adopt a specific time frame.

To identify patterns of the relationship between short and long channels, the municipalities were categorized into (1) with many long and short channels, (2) with few long and short channels, (3) with many long and few short channels, and (4) with many short and few long channels. The cutoff point to define what was too much or too little was half the number of channels verified in the municipality with the most. Twelve short channels and seven long channels were the maxima verified in the surveyed data. Therefore, municipalities with many long channels were considered four to seven; municipalities with few long channels, one to three; municipalities with many short channels, six to twelve; and municipalities with few short channels, one to five. This categorization served to identify possibilities for statistical analysis.

To compare the types of short and long channels in the state of Goiás, the quantity of each type was compared in a paired manner in each municipality using a paired *t*-test with 95% significance. A box plot graph representing error deviations, standard deviations, and mean was plotted to represent and visualize the mean values.

From the indications obtained by this categorization, the Euclidean Distance tool was used to verify grouping patterns. The inverse or complement of similarity is distance measures. Thus, the distance between the different short- and long-channel modalities verifies the dissimilarity between these different channels. The Euclidean distance uses the unit of each channel type to calculate the rejected distance between them, using the Pythagorean theorem, thus generating the difference measures between the groups. To verify the dissimilarity between the set of long and short channels among the municipalities, the Euclidean distance with a bootstrap of 10,000 repetitions was used. A cluster analysis was performed using the variable number of family farmers participating in each channel. The analysis was visually represented with a dendrogram, prioritizing the groups formed by nodes above 62. The graphical representation of the Euclidean distance in the form of a dendrogram visually demonstrates the groups based on the geometric distance between the groups, not presenting another measurement unit other than the distance itself. From the analysis of the dendrograms, we proceeded to the discussion about the types of coexistence existing between the channels. Using as reference the typification proposed by Thomé et al. [8], however, we obtained three types of coexistence with some modifications: first, inclusive—large participation of family farmers in one channel promotes their inclusion in others; second, excluding—large participation of family farmers in one channel promotes their exclusion in other channels; third, neutral—there are no clear relations of inclusion or exclusion of family farmers in any channel due to their large participation in another channel.

Maps were prepared using a geographic information system (GIS) using Qgis software (2.18). The data collected during the survey were related to the Goiás State Government Geoinformation System (SIEG) database. The maps present the level of diversity of short and long channels with family farmer participation in the sampled municipalities.

## 4. Results

Table 4 presents the average number of short channels (SC) and long channels (LC) in the studied municipalities, the differences in the means, and their respective standard deviations and errors. High standard deviations are noted for both short and long channels.

**Table 4.** Occurrence of short channels (CC) and long channels (CL) in Goiás in 2019. Source: survey data collected in 2020 (different letters correspond to statistical differences using *t*-test with 95% confidence).

| | Average | | Standard Deviation | | Standard Error | |
|---|---|---|---|---|---|---|
| | **SC** | **LC** | **SC** | **LC** | **SC** | **LC** |
| Number of short channels available to family farmers | 7.18a | 2.95a | 2.49 | 1.29 | 0.20 | 0.10 |
| Number of commercialization outlets occupied by family farmers | 121.10a | 195.94b | 185.57 | 261.60 | 14.90 | 21.01 |

Table 4 also reveals that although short channels are more diverse, long channels are responsible for creating more marketing outlets for family farmers. The total number of short-channel marketing outlets taken by family farmers in the 155 sampled municipalities was 18,771, compared to 30,371 long-channel outlets. Out of the marketing outlets occupied by family farmers, 61.8% are of the long channel type while 38.2% are of the short channel type. When calculating the inclusion index of the different modalities, through the ratio between the number of marketing outlets and the number of rural family households in the sampled municipalities, 71,504 households, the short channels present an inclusion index of 26.25%. In contrast, the long channels present an inclusion index of 42.47%. Therefore, 67.73% of the sampled family farmers participate in some commercialization channels, and 31.28% do not.

Figure 4 shows a boxplot that illustrates the variation in the average marketing modalities of the short and long channels. Each municipality has, on average, 2.93 long and 7.17 short channels, while t = −23.84 and *p* = 0.000. The minimum amount of short channels in the sampled municipalities is 1, while in the long channels, it is 0. The maximum values reveal that the municipalities with more long channels have seven modalities, while the municipalities with more short channels have twelve. The most frequent number of long channels among the sampled municipalities was between two and four modalities. Among the short channels, the most frequent quantity ranges between five and nine commercialization modalities.

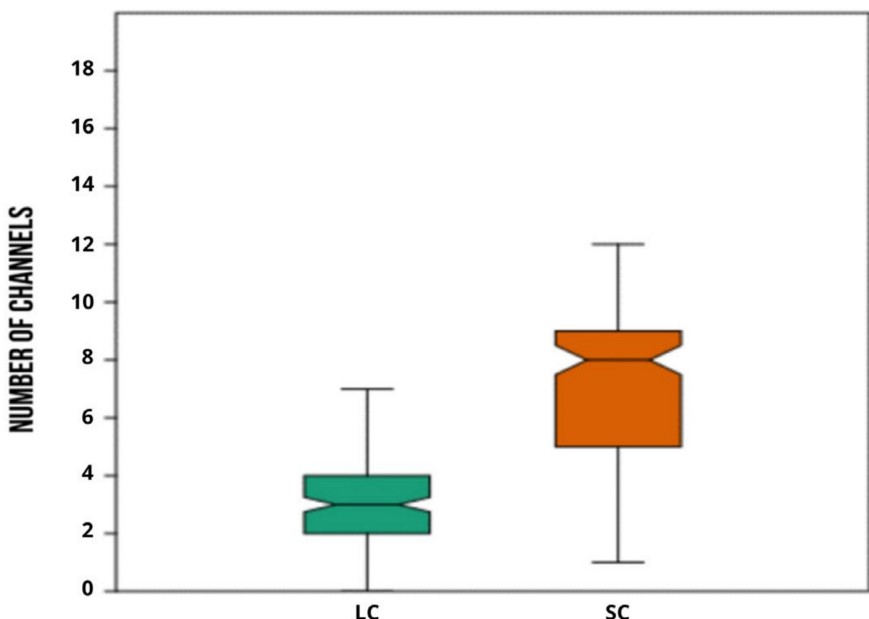

**Figure 4.** Boxplot—comparison between the number of short- and long-channel modalities with family farmers' participation in the State of Goiás. The green boxplot refers to the number of long channels and the orange boxplot to the number of short channels. Source: survey data collected in 2020.

Table 5 reveals that, except for the long channel of milk sales to dairy industries, no marketing channel can include a large portion of family farmers. The participation rate of family farmers in each channel was estimated at 70.76%. Therefore, it is possible to infer that 31.28% of family farmers in the sampled municipalities do not participate in commercialization.

**Table 5.** Proportion of family farmers included in each marketing channel. Source: survey data collected in 2020.

| Short Channels | | Long Channels | |
|---|---|---|---|
| **Channel** | **FF on Channel (%)** | **Channel** | **FF on the Channel (%)** |
| Neighbors | 5.99 | Milk to dairy | 27.58 |
| Fair | 5.62 | Slaughterhouses | 11.79 |
| Local butchers | 3.22 | Agricultural commodities | 4.15 |
| Institutional market | 2.25 | Warehouses | 1.91 |
| Door to door | 1.71 | Long-channel extractivism | 1.25 |
| Sale on property | 1.92 | Integration systems | 0.23 |
| Apps and social networks | 1.70 | Other long channels | 0.7 |
| Short-channel extractivism | 1.09 | | |
| Stalls on city streets | 0.50 | | |
| Innovation—new channels | 0.52 | | |
| FF specific store | 0.20 | | |
| Basket system | 0.14 | | |

Most municipalities included in the sample have either many long channels and short channels or few short channels and few long channels (50.26%). Many municipalities have many short channels but few long channels (47.15%), and a minority have many long channels and few short channels (2.59%).

The cluster analysis with the number of family farmers participating in long and short channels (Figure 5) has shown not only the formation of some clusters but also sets of channels. The variables of each group are described in Table A1 in Appendix A (N—number of locations that have the commercialization channel; Max—the maximum number of locations that have the commercialization channel, Sum—sum of locations that have the commercialization channel, Mean—average of locations that present the commercialization channel; Std. Error—error deviation; variance—variance; Stand. Dev—standard deviation of the locations that present the commercialization channel). The three main long channels are categorized together as the LC Set and are milk sales to dairies, defined as milk; cattle sales to slaughterhouses, defined as slaughterhouse; and sales of agricultural commodities, defined as commodities. They are not related to the larger group referred to as SC and LC Set 1. One can notice the formation of a grouping that extends from the warehouse channel, defined as a warehouse, to the sales channel to PNAE (public procurement), defined as procurement, present in the SC and LC Set 1 categorization.

The SC Set 2, made of the channels sales at fairs, defined by fair, and sales to neighbors, defined by neighbors, does not cluster with any specific channel but relates to the SC and LC Set 1 and several short and long channels. Also forming clusters were SC Set 2, consisting of the channels door-to-door sales, defined by door, and sales to local markets, defined by l. market, and SC and LC Set 2, consisting of the long channel sales to procurement and the short channel innovation/new markets, defined by innovation.

Figure 6 depicts marketing channels with family farmers' participation (map A—short channels; map B—long channels). A congruence is observed between municipalities with the high or intermediate occurrence of short channels—8 to 12 and 4 to 7 channels, respectively—and high or intermediate occurrence of long channels—5 to 7 and 3 to 4 channels, respectively. Very few municipalities with a high occurrence of long channels do not also present a high occurrence of short channels. However, most municipalities that present a high occurrence of short channels do not present a high occurrence of long channels. The state's northeast region presents a cluster of municipalities with a high occurrence of short channels and a low or intermediate occurrence of long channels. In the southern portion of the state, most municipalities presented a high occurrence of long channels. Many municipalities in this region also presented a high occurrence of short channels. It is noted that the moderate or high occurrence of short channels is more frequent

and more evenly distributed than the moderate or high occurrence of long channels. No trend suggesting causal relationships has been identified. That is, it cannot be said that the high occurrence of long channels is related to the low occurrence of short channels and vice versa.

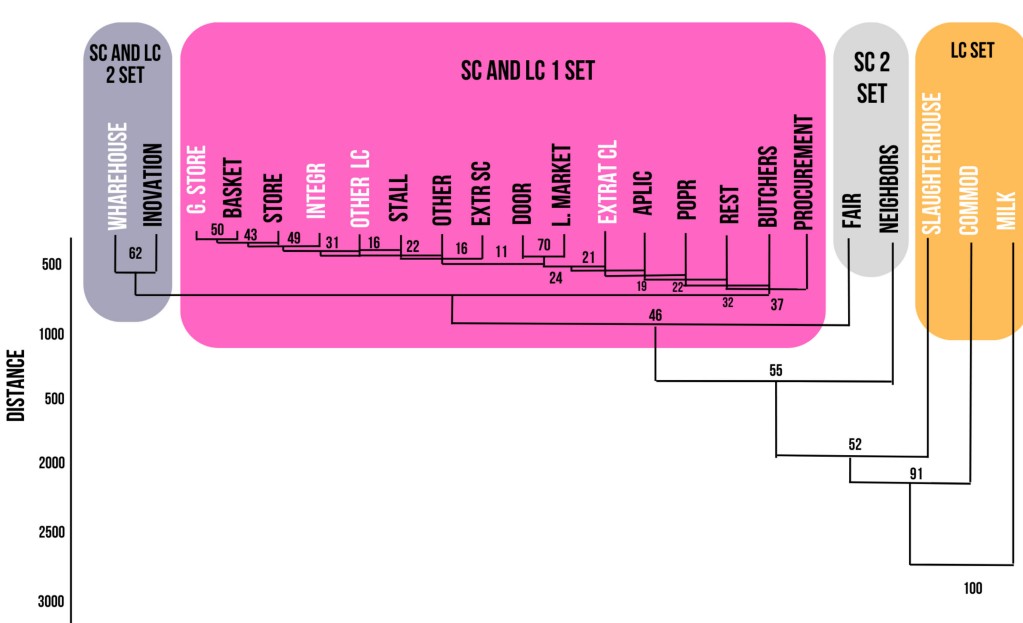

**Figure 5.** Euclidean distance dendrogram used to group by dissimilarity the number of family farmers who participate in the short- and long-channel modalities in the State of Goiás. Source: survey data collected in 2020.

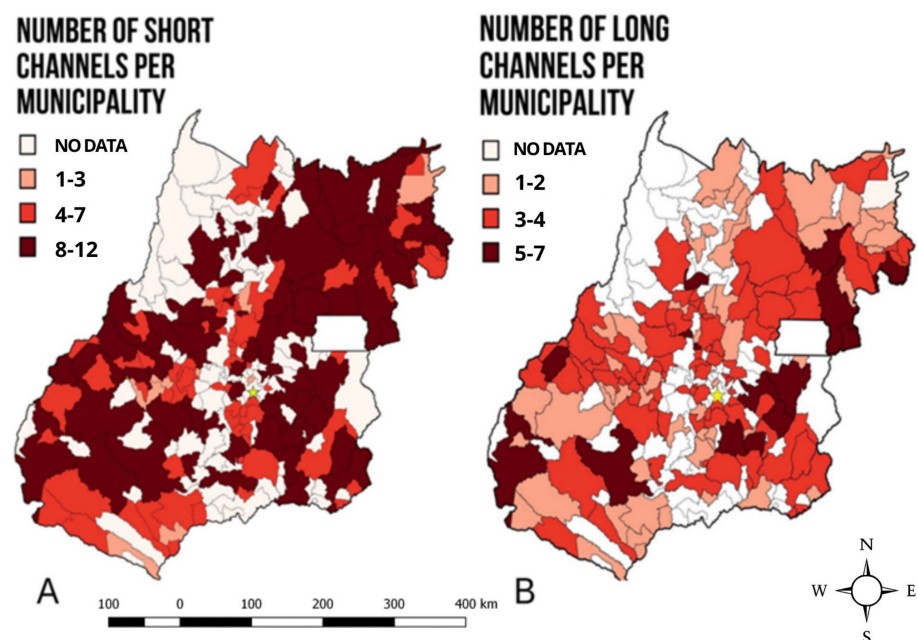

**Figure 6.** Diversity of short and long channels with family farmers' participation in the municipalities of Goiás. The intensity of the color indicates the number of channels per municipality. (Map (**A**) represents short channels and map (**B**) long marketing channels). Source: survey data collected in 2020 and SIRGAS 2000.

The income obtained by the sampled family farmers is from sales via long channels (74.26%), but short channels also generate significant income for this category (25.74%). The long channels that generate the most income are those linked to the cattle production chain, and among the short channels, the only one that generates significant income is the fair channel (13.24%). Table 6 presents the three main short channels and the three main long channels regarding income generation.

**Table 6.** Main short and long channels for income generation by the sampled family farmers. Source: survey data collected in 2020.

| Short Channels | | Long Channels | |
|---|---|---|---|
| Channel | Income Generation (%) | Channel | Income Generation (%) |
| Fair | 13.24 | Milk to dairy | 32.82% |
| Local butchers | 2.75 | Slaughterhouses | 11.79 |
| Local markets | 2.34 | Integration systems | 10.14 |

## 5. Discussion

The low occurrence of short and long marketing channels hinders the development of local food systems where family farmers can market their goods [9]. One-quarter of the municipalities surveyed in this study have only few short and long channels. Therefore, these municipalities are more susceptible to problems with food supply. The low diversity of marketing channels with the participation of family farmers may also be related to the low local income level [1], which directly reflects the farmers' consumption level [2].

On the other hand, about one-quarter of the sampled municipalities present a high occurrence of both short and long channels. This pattern may be related to the coexistence of actors and organizations that act in both long and short channels, generating greater resilience of the agri-food system in these municipalities [11]. Therefore, the model based on short channels does not operate in isolation but connects with the model based on long channels [7].

The context of dispute and tensions between the different models of food production and distribution and the history of family-based agriculture in Brazil, marked by processes of low access to public policies, cast doubt on the harmonious coexistence between models based on short and long channels [14]. The simultaneous operation of the long-channel model has consequences for the short-channel model and vice versa. This can be verified by the low participation rates of family farmers in the short and long channels and by the high standard deviations and standard errors of the quantity of trading outlets of the short and long channel types.

There is a coexistence of models; however, none can effectively include family farmers, corroborating the hypothesis of false harmonic coexistence [14]. Long and short channels cannot include the entire category of family farmers since not all family farmers fit the model based on long channels and not all family farmers can access the short channels.

The fact that short channels include few family farmers is not necessarily a negative aspect. Short-channel experiences that grow a lot in the number of consumers and farmers tend to approach the logic of the long-channel agri-food model, losing the characteristics of short channels [31]. This is a debate with several nuances because short channels have weaknesses that are strengths in long channels, such as production scale and supply chain coordination. The literature and the social and productive organizations in the field have sought to answer how and to what extent the increase in production scale and the qualification of chain coordination in short channels can occur without losing their quality attributes [31].

The large occurrence of municipalities with a great diversity of short channels may be related to the fact that the initiatives in Goiás are connected to the fudging characteristics of short channels. Nonetheless, the standard deviation of the average number of short channels is very high, showing the occurrence of some municipalities with very few

modalities of this type of commercialization. Short channels with many family farmers tend to lose the profile and identity of a short channel and approach the logic of a long channel. Therefore, the low percentage of family farmers in short channels is offset by the great diversity of channels.

Long channels may create inter-network connections that contribute to the professionalization of the agents and the qualification of the institutions' organization. Short channels can create intra-network connections that establish links of trust between farmers and consumers. Inter-network and intra-network connections are not antagonistic and can occur in the same territory [12]. The data reveal that the high occurrence of long channels does not prevent the emergence and consolidation of large amounts of short channels and vice versa. On the other hand, the low occurrence of one of the channel types seems to be related to the low occurrence of the other type; that is, what causes the channels to emerge and consolidate is not the existence of some other channel but rather the contexts and characteristics of the territories and their agents. These interactions will condition the type of existing coexistence between the different models of food production, distribution, and consumption [32].

Some farmers participate in more than one commercialization channel and move between short and long channels, according to their profile and the characteristics of the agri-food system [3]. This study reveals that some municipalities have high inclusion rates of family farmers in both short and long channels.

The typification of coexistence—inclusive, excluding, and neutral—between different models of food production and distribution is useful for identifying possibilities of synergies between actors and organizations in agri-food systems [8]. The clusters obtained through the dendrogram suggest some relationships between long and short channels. The formation of clustering between most of the short and long channels—Set of CC and CL 1, in Figure 3—suggests a neutral coexistence relationship between most short and long channels. That is, the increased participation of family farmers in short channels in this cluster is not related to the lower participation of these farmers in the long channels of the same cluster, and vice versa.

The cluster analysis formed groups that can be used to differentiate the profile of each channel modality. Nevertheless, since the data collection was carried out in a specific year, it reflects family farmers' participation in the marketing channels at a specific moment. Knowing that the market inclusion/exclusion of these farmers is a process, it would be interesting for future studies to use cluster analysis based on time-series data.

A neutral coexistence verified at a given moment may not be sustained over time [26] because one of the dimensions of coexistence between agri-food models is the tension between diversification, more recurrent in short channels, and specialization, markedly present in long channels [9]. The tension between diversification and specialization in production, distribution, and consumption is translated into the conflict between actors and organizations that operate in short and long channels and shape institutions, such as sanitary legislation, technical profiles of agricultural science professionals, and public policies, which historically favor the agri-food model based on long channels [13].

The fact that coexistence is neutral does not indicate any relationships between the channels. Many farmers participate in several marketing channels, and often, participation in one channel drives a farmer's entry into another. It is noted that some farmers who started their sales in restaurants and snack bars or on their property—directly to the consumer—see a new opportunity in sales through apps and social networks and start marketing their products in this new channel. With the opening of new contacts, these farmers articulate themselves to create a collective initiative, such as creating specific family farmer stores or basket systems. Depending on their economic and infrastructure situation, some extractivist farmers sell their products, sometimes directly to the final consumer, sometimes to intermediaries. Some farmers who sell their production via PNAE specialize in certain products, become trained in production sizing and scheduling, and start selling to local markets or supermarket chains.



The long channels selling milk to dairies, cattle to slaughterhouses, and agricultural commodities—CL cluster in Figure 3—are not directly grouped with the other clusters, suggesting an exclusive coexistence relationship between the channels in this cluster and the others. All three channels are focused on a single productive activity, which hinders diversification. The cultivation of agricultural commodities—soybean, corn, cotton—and raising beef cattle, to some extent, demand large extensions of land, and the sale of milk to dairies presents a relatively high demand for labor. As one of the major bottlenecks for most Brazilian family farmers is the low availability of land and labor [33], the participation of these farmers in one of these channels may be related to their low participation in others. These channels exclude coexistence with the other channels since they exclude the possibility of family farmers seeking new markets. Thus, the promotion of the resilience of local agri-food systems and the construction of an articulated coexistence between short and long channels [12] are weakened when there is a large participation of family farmers in marketing channels—milk sales to dairies, cattle sales to slaughterhouses, and sales of agricultural commodities.

Other studies have highlighted the consequences of family farmers' participation in these three long channels, including the changes in the configuration of labor and population flows of rural populations when there is a large insertion of family farmers in soybean monocultures [15], loss of identity and personalization of the production of family farmers when they no longer sell their milk production through short channels and insert themselves in long channels [37], and decrease in product diversification and loss of autonomy of rural communities when farmers start to dedicate themselves primarily to monocultures [38].

The coexistence of distinct models of production, distribution, and consumption in Brazil is marked by disputes over political hegemony and greater support from the state [14]. This dispute in Brazil and other countries has materialized in creating norms, legislations, and institutions that favor model development based on long channels [13,14]. Large companies and their representative organizations have a greater presence and prominence in long channels. However, in short channels, there is the lead of consumers, farmers, and their relationships [8]. Thus, it is possible to infer that the political action of companies and organizations representing the production chains linked to the long channels is related to the exclusive coexistence between the three main long channels and the other marketing channels. Farmers feel more secure in these channels because the chains are more structured, and the institutional arrangement, both state and private, is clearer. However, the need for specialization and productive intensification, a characteristic of the three main long channels studied, makes it infeasible for family farmers to insert themselves in other marketing modalities.

The short channels selling at fairs and selling to neighbors—Set of CC 1, in Figure 3—are outside the large grouping formed—Set of CC and CL 1, in Figure 3—suggesting that there is no relationship between these channels and any other one specifically, but rather with the set of channels. Fairs are the main sources of income for family farmers via short channels [39], and sales to neighbors are a neglected but recurrent marketing modality. These channels do not operate in isolation. Fairs, especially, are spaces where farmers articulate socially and politically and professionalize themselves in both productive and commercial aspects. Therefore, the large participation of family farmers in fairs generates economic developments capable of generating opportunities for this category in other marketing channels. Thus, the channels of CC Set 2 present a relationship of inclusive coexistence with the channels of CC Set and CL 1.

In the state of Goiás, the difference between the south and the north is noticeable in several aspects. The southern part presents the largest production of grains and other commodities, the highest levels of consumption of pesticides and acquisition of agricultural machinery, and the best transportation infrastructure. On the other hand, the northern portion has the largest number of squatter families and the highest rates of violence and vulnerabilities of family-based farmers [33]. Such differences between the south and the

north are translated into the configurations and spatial relations of the agri-food systems of the municipalities that make up each region.

The maps revealed that, in the southern portion of the state, there is some congruence between municipalities with a high occurrence of short channels and a high occurrence of long channels. They also revealed that in the northern portion, such congruence does not exist. Economically dynamic regions favor the emergence and consolidation of distinct production, distribution, and consumption models, suggesting the coexistence of short and long channels. Although no studies establish a causal relationship between the high occurrence of short and long channels and local economic dynamism, regions with higher levels of food security tend to present inclusive and cooperative coexistence relations between short and long channels [11]. Short channels are diversified in economically less dynamic regions with impoverished populations since economic flows are more intense within territories than between them. Therefore, in these regions, the short channels are primarily responsible for the food security of the municipalities.

The data on income generation are not very conclusive as inferential statistical treatment was not carried out. However, results reveal that the long channels generate most of the income for family farmers and that the short channels are emerging as an alternative for the commercialization of products by family farmers. Issues related to production and transaction costs in the different marketing channels were not investigated. Future studies may adopt the scope of the present research (relationship between short and long channels), however, using inferential statistical methods such as those used by Briones (2015) [30].

The results present an approach that goes beyond the simple dichotomy between long and short channels and beyond the inevitable and harmonious convergence between them. The nuances and specificities of each channel and reality create complementarities and antagonisms because their borders are "permeable" and constantly disputed. It is also worth pointing out that coexistence is perceived not only among marketing channels or agri-food models but also among agents and products. Farmers can sell cheeses, for example, in the formal market—with inspection seals—to supermarket chains and at fairs or on their property directly to consumers [16].

## 6. Conclusions

This study adds a quantitative assessment to the existing mostly qualitative literature on family farmers' market access. This study sought to assess how the coexistence between short and long marketing channels promotes or inhibits family farmers' inclusion in different markets. The results revealed exclusive, neutral, and inclusive coexistence relationships between long and short marketing channels.

Both short and long marketing channels in the Brazilian state of Goiás have low inclusion rates of family farmers since 31.28% of family farmers did not participate in any marketing modality. Short channels offered more diversity of marketing possibilities than long channels, with an average of 7.18 types of short channels and 2.95 types of long channels available in each sampled municipality. However, long channels provided more outlets for commercialization than short channels. Among the 49,142 commercialization outlets occupied by the surveyed family farmers, 38.20% were short channel types while 61.80% were long channel types.

Municipalities with a great diversity of short channels tended to have a great diversity of long channels, and vice versa. Even if including fewer family farmers, the short channels presented the greatest diversity of possibilities for commercial inclusion in most municipalities in Goiás.

Inclusive and exclusive coexistence relations were found between some modalities of short and long channels. Greater participation of family farmers in long channels for the commercialization of a single product—such as agricultural commodities and cattle—was related to low participation in the other channels, which revealed exclusive coexistence between these channels.

Recognizing that long channels based on a single product inhibit family farmers' participation in other marketing channels is essential to understand the agri-food systems bottlenecks. By understanding these bottlenecks, actors linked to food production (public authorities and civil society), distribution, and consumption and public policymakers will be able to plan strategies to promote commercialization channels that contribute to increasing local food security.

The theoretical and methodological limitations of this study are as follows: (1) the relationships between the agents that operate in the different channels were not analyzed; (2) no data were collected that characterize concepts and qualitative attributes of agri-food systems such as hegemony, diversification, specialization, innovation, and adaptation; (3) the large scope of the present study made an in-depth investigation of the aspects mentioned above unfeasible; (4) the fact that the interviews were carried out remotely (due the COVID-19 pandemic) prevented the perception of nuances that could be valuable in characterizing the relations of coexistence between the agri-food models; and (5) studies on short channels are recent in Brazil and not frequent in Goiás, and the academic debate on the coexistence of agri-food models in Brazil is still incipient, making this research lack local theoretical bases.

**Author Contributions:** Conceptualization, T.d.C.V.; methodology, T.d.C.V.; investigation, T.d.C.V.; data curation, T.d.C.V.; analyses and draft preparation, T.d.C.V.; writing—review and editing, T.d.C.V.; literature review, T.d.C.V.; methodology, C.d.M.e.S.N.; data curation, C.d.M.e.S.N.; writing—review and editing, C.d.M.e.S.N.; literature review, C.d.M.e.S.N.; methodology, G.d.S.M.; data curation, G.d.S.M.; writing—review and editing, G.d.S.M.; literature review, G.d.S.M. All authors have read and agreed to the published version of the manuscript.

**Funding:** This research received no external funding.

**Data Availability Statement:** Data presented in this study are available upon request to the author for correspondence. The data have yet to be publicly available, as they will be used in future studies submitted to journals that require originality.

**Acknowledgments:** We would like to thank the Coordination for the Improvement of Higher Education Personnel Foundation (CAPES) for granting a PhD scholarship. Without such support, this research would not be possible.

**Conflicts of Interest:** The authors declare no conflict of interest.

## Appendix A

**Table A1.** Variables of each marketing channel group.

|  | N | Max | Sum | Mean | Std. Error | Variance | Stand. Dev |
|---|---|---|---|---|---|---|---|
| MILK TO DAIRY | 133 | 1430 | 17,774 | 133.6391 | 15.77624 | 33,102.35 | 181.9405 |
| WAREHOUSE | 58 | 300 | 1293 | 22.2931 | 5.851717 | 1986.07 | 44.56535 |
| SLAUGHTERHOUSE | 97 | 876 | 7010 | 72.26804 | 12.37451 | 14,853.47 | 121.8748 |
| SUPERMARKET CHAIN | 11 | 20 | 73 | 6.636364 | 2.090118 | 48.05455 | 6.932139 |
| AGRICULTURAL COMMODITIES | 52 | 1515 | 2867 | 55.13462 | 29.11036 | 44,065.49 | 209.9178 |
| INTEGRATION SYSTEMS | 13 | 50 | 145 | 11.15385 | 4.205988 | 229.9744 | 15.16491 |
| LONG-CHANNEL EXTRACTIVISM | 37 | 75 | 707 | 19.10811 | 3.572245 | 472.1547 | 21.72912 |
| OTHER LONG CHANNELS | 14 | 80 | 264 | 18.85714 | 5.762571 | 464.9011 | 21.56157 |
| FAIR | 124 | 250 | 4015 | 32.37903 | 3.752656 | 1746.221 | 41.78781 |
| INSTITUTIONAL MARKET | 155 | 201 | 1675 | 10.80645 | 1.997508 | 618.4558 | 24.86877 |
| NEIGHBORS | 89 | 840 | 3401 | 38.21348 | 10.29559 | 9433.92 | 97.12837 |

**Table A1.** *Cont.*

|  | N | Max | Sum | Mean | Std. Error | Variance | Stand. Dev |
|---|---|---|---|---|---|---|---|
| SALES TO RESTAURANTS AND SNACKS | 98 | 250 | 1289 | 13.15306 | 2.694773 | 711.6567 | 26.6769 |
| BASKET SYSTEMS | 15 | 30 | 99 | 6.6 | 1.973153 | 58.4 | 7.641989 |
| FAMILY FARMER SPECIFIC STORE | 18 | 25 | 145 | 8.055556 | 1.867283 | 62,76144 | 7.922212 |
| DOOR TO DOOR | 106 | 60 | 1097 | 10.34906 | 1.189052 | 149.8675 | 12.24204 |
| SALES ON PROPERTY | 82 | 150 | 1254 | 15.29268 | 2.772596 | 630.3577 | 25.10693 |
| APPS AND SOCIAL NETWORKS | 68 | 120 | 1079 | 15.86765 | 2.851528 | 552.9225 | 23.5143 |
| LOCAL MARKETS | 122 | 80 | 1274 | 10.44262 | 1.129535 | 155.6537 | 12.47613 |
| SHORT-CHANNEL EXTRACTIVISM | 59 | 76 | 638 | 10.81356 | 1.828164 | 197.1888 | 14.04239 |
| STALLS ON CITY STREETS | 42 | 100 | 290 | 6.904762 | 2.391382 | 240.1858 | 15.49793 |
| INNOVATION—NEW CHANNELS | 7 | 340 | 372 | 53.14286 | 47.92625 | 16,078.48 | 126.8009 |
| OTHER SHORT CHANNELS | 27 | 65 | 420 | 15.55556 | 3.324061 | 298.3333 | 17.27233 |
| LOCAL BUTCHERS | 64 | 120 | 1652 | 25.8125 | 3.71277 | 882.2183 | 29.70216 |

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
