# Peer review of "Family Farmers in Short and Long Marketing Channels: Lessons for Rural Development in Goiás, Brazil"

_logistics, 2023_

Round 1

Reviewer 1 Report

‘Family farmers in local and commodities markets’.

This paper introduces an important topic for logistics and fits well in the general purpose of the journal. The objective of the paper and the problem statement are not specified in the abstract.

Despite of the interest of the topic of the article, this proposal has many weaknesses: the abstract is poorly written, some concepts are not clearly defined, the introduction is mainly descriptive, there literature review is weak, the methods should be detailed and improved, external validity was not implemented, the conclusion should be completed and completely reframed…

The title of the paper suggests it is quite ambitious as it intends to provide an examination of the ‘family farmers in local and commodity markets’.

The title should be narrowed and aligned with the problem statement and with the literature. Suggestion: maybe you should add a sub-title referring to short supply chains and the territory concerned by the study.

The abstract should clearly specify the main problem statement and the main streams of the literature review.

The list of keywords is incomplete. For example, the words ‘distribution’ and ‘marketing channels’ or ‘short supply chains’, ‘cattle’, and ‘Brazil’ should be included.

The general introduction is mainly descriptive. Please make it more explanatory. Explain the theoretical, practical, and methodological interests of this paper.

Some abbreviations are not explained (example: CEASA, page 1).

At the end of the Introduction, we expect to see a presentation of the structure of the different parts of the paper (Part 1, Part 2, etc).

It is very surprising to see the section dedicated to the Literature Review after the Methods. This is completely inappropriate.

The methods should be summarized and presented in a Figure where we can identify the different steps establishing the questionnaire/interview guides and the data collection, and sampling process.

There is not a detailed presentation of the variables introduced in the question and a discussion of the measurement instruments.

There is no discussion about the importance and appropriate use of Venn diagrams. This seems to be a basic tool.

The literature review should be placed before the methods. The literature review should be streamlined and refocused on the literature on short supply chains or marketing channels.

The word ‘coexistence’ was repeated too many times (56 times in the paper). Overuse of some words or sentences in English denotes poor writing. The sentence ‘coexistence of agri-food channels in Brazil…’ was repeated in two consecutive paragraphs (lines 281, 291).

In some cases you used the expression ‘actors’ and in others ‘players’ (line 250). Does this refers to the same thing ?

The style of the writing should be improved. Many expressions are not appropriate for a scientific work. For example:

*what is a ‘critical eye’ (line 277) ?

*’The coexistence of agri-food models in Brazil was never peaceful’ (line 291). What ‘peaceful’ means ?

*’the distinction between…. Is clear’. (lines 305-306). It is clear to whom ?

The limitations of the study should be included in the conclusion (lines 315 and following).

We cannot link the 12 modes in each municipality to table 1 (see lines 330-331).

Some tables are incomplete. For example, what are the units in Figure 3 ? (km ?)

Figure 4 presents two maps. What is the scale of those maps ? The word ‘subtitle’ should be removed.

The general discussion should be streamlined and references to the literature review are required. The discussion refers to facts and outputs of the survey but it lacks connections with the scientific literature review.

Further, the discussion should compare your findings with the findings in the literature review.

I recommend that you present a table introducing the different dimensions and your findings.

Some sentences are vague (example: lines 586-587).

The limitations of the study should be added in the conclusions, not in other parts of the paper (lines 513-522). The potential contribution(s) of the study should be highlighted in the conclusions, not in the section dedicated to the discussion (lines 557-560; lines 609-616).

The general conclusion should start by reminding what the initial problem statement is and the main answers provided by the study.

What are the key theoretical, practical, and methodological contributions of this paper ?

What are the theoretical and methodological limitations of the paper? Nothing is written about the limitations of materials and methods. Limitations about the instruments to collect data ?

Any extensions related to potential new theoretical contributions?

The references included in the paper should be extended to the main international publications related to short supply chains and emerging markets.

Many other parts of the paper require fine tuning.

In summary, paper should be entirely rewritten and reconsidered.

A major rewriting is required: style, abstract, structure of the sections, literature review, methods, presentation of the results, discussion, and conclusions.

The style of the writing should be improved. Senteces should across the paper should be streamlined. There are too many repetitions.

Many expressions are not appropriate for a scientific work.

For example:

*what is a ‘critical eye’ (line 277) ?

*’The coexistence of agri-food models in Brazil was never peaceful’ (line 291). What ‘peaceful’ means ?

*’the distinction between…. is clear’. (lines 305-306). It is clear to whom ?

Author Response

Thanks for de valuable suggestions. I hope that with de adaptations made, the manuscript will have better quality. 
The suggestions are answered point by point in the attached table.

Reviewer 2 Report

This study provides an interesting topic and appropriate topic for family farmers' access to market in the Brazilian state of Goiás through the application of questionnaires and interviews conducted with rural outreach agents and family farmers’ leaders.

This study provides an interesting topic and appropriate topic for family farmers' access to market in the Brazilian state of Goiás through the application of questionnaires and interviews conducted with rural outreach agents and family farmers’ leaders.

Author Response

Thanks for the valuable comments. I hope that with the adjustments made, the manuscript will reach higher quality The suggestions are answered point by point in the attached table.

Round 2

Reviewer 1 Report

‘Family farmers in short and long food supplychains: lessons for rural development’.

This paper introduces an important topic for entrepreneurship studies and fits well in the general purpose of the journal. Following the abstract, the objective of the paper is ‘to assess to what extent the relationship between short and long-marketing channels promotes or inhibits the commercial inclusion of family farmers’.

Despite of the focus of the general purpose of the article, this proposal has many weaknesses: the title is not aligned with the contents, some concepts are not clearly defined, the abstract should be rewritten, the introduction is mainly descriptive, there is limited literature review, the methods should be detailed and improved, external validity was not implemented, the conclusion should be completed and completely reframed…

The title of the paper suggests it is quite ambitious as it intends to provide an examination of the ‘family farmers in short and long food supply chains’.

The title and/or the contents of the paper should be aligned. In the body of the text the authors use multiple expressions but the dominant ones are not, by far, the ones quoted in the title of the paper: chain (quoted only 5 times), supply (11 times), channels (quoted 282 times), channel (68 times), marketing channels (2 times), marketing channel (6 times), commercialization (12 times), marketing (52 times), and farmers (122 times). The analysis of the most quoted expression in the body of the text is ‘channels’, not ‘supply chains’. Therefore there seems to be a misfit between the title of the paper and what is developed in the body of the paper. We suggest that the authors define the concepts and harmonize the expressions in the body of the paper and align the title.

Further, there is a dedicated and mainstream literature on global value chains (GVC) and economic development impacting farmers in developed and developing countries (Gereffi and Korzeniewicz, 1994; Gereffi, 1996; Gereffi et al., 2001; Gereffi, 2005; Humphrey; Gereffi and Lee, 2012, Humphrey and Memedovic, 2006; etc). Why such literature is not taken into consideration in the context of the current paper?

The abstract indicates the study intends to assess the relationships between short and long supply chains to promote or inhibit commercial inclusion. However, in the body of the text we also see a discussion and empirical data related to ‘vulnerability’.

The sentence in lines 43 and 44 is not complete (links between agents and ???).

How do you define what an ‘agent’ is ?

In the general introduction please indicate what refers to ‘self-consumption’ of farmers’ production (see, for example, lines 66-67).

The authors justify the interest of the paper (‘research gap’) is based on the lack of ‘precise information in the Agricultural Censuses on how the products produced by family farmers are commercialized, and also, the scientific literature lacks quantitative studies that evaluate the economic relevance between short and long channels’ (lines 74-78).

The interests of a paper can be theoretical, practical, and methodological. However, in a science-based paper you are expected to use/introduce one or several research streams to justify your problem statement.

This statement related to the unavailability of data can be (partly) accepted for a research paper however you will need to make a literature review of the current literature review and summarize it on a paper. Such statement was not demonstrated in the general introduction or in the section dedicated to the literature review. Please summarize the literature review with previous studies as well as the orientation of the studies (qualitative/quantitative, mixed-methods…) on a table.

At the end of the introduction, you should present thoroughly the organization and the contents of the sections included in the paper.

The section dedicated to the literature review includes some general expressions/statements without explaining or defining such concepts. For example, the paragraph in lines 94-100 is merely general/broad.

The concept of ‘food orders’ (line 101) is not defined. The sentence referring to the ‘concept of food orders’… ‘takes into consideration the coexistence of several food orders’ (lines 104-106) does not clarifies what a ‘food order’ is.

The ‘dichotomy’ between short and long chains is repeated multiple times (in the general introduction, in the literature review, etc). (lines 22-23, 151, 199, 248-249…). The word ‘coexistence’ is repeated 56 times (!) in the body of the text. Please avoid repetitions in the body of the text.

Can you summarize on a figure the literature/concepts/comparison related to short and long channels ? (supply chains ?). What are the differences summarizing the different ‘models’ ? (line 115).

How do you define what is an ‘inclusive’ and ‘vulnerable’ food system ?. Is a ‘food system’ the same as a ‘food channel’ ? (line 119) and ‘agri-food system’ (line 127) ?, ‘agri-food models’ (line 184)..

Please avoid using normative statements in the body of the text. For example, ‘the search for more inclusive and less vulnerable food systems is fundamental…’ (line 119). ‘This inclusive character can be achieved by combining…’ (line 120). These are normative statements, not appropriate for a research paper. Further, such statements were not demonstrated in your literature review or empirical study.

How do you define what is ‘the identity’ of a short channel ? (line 132).

After reading the paper, it is difficult to understand what is the conceptual positioning and the theoretical streams developed in the paper. There is no research question and therefore we cannot link it to the methods and to further discussions.

The ‘keywords’ presented in the paper are very limited and do not represent the essence of the paper. For example, the two following keywords summarize a similar focus: ‘entrepreneur’ and ‘entrepreneurship’. Other possible keywords could include: ‘Brazil’, ‘survey’, etc.

The section dedicated to the literature review should be entirely rewritten. The concepts should be defined; the distinction between short/long chains should be clearly made (build also a figure explaining the dimensions involved, previous studies, etc). The literature review should identify the research gap and to help you to build a conceptual framework to build bridges with the empirical study to be developed in the methods. In the general introduction your research ‘gap’ was mainly defined in relationship with the lack of quantities studies. This argument was not clearly established and discussed in the section dedicated to the literature review. What were the qualitative, quantitative studies and other studies involving mixed-methods that related to the study of short/long supply chains.

In the general conclusion, the authors wrote that the study ‘ contributes to the literature by revealing that the consequences of the large participation of family farmers in long channels focused on mono-crop production impacts not only the category of family farmers but also the promotion of resilient agri-food systems’. (lins 664-667). There was not survey of the literature review to discuss the consequences of the participation of family and non-family farmers in the agri-food chains or resilient food systems. In the absence of a systematic review of the literature, the authors cannot advance such conclusions. Further, contrary to a research gap on ‘quantitative studies’ this point was not highlighted in the general introduction as a potential contribution of the paper

Why in the conclusion you remind again that your study ‘refutes the dichotomy between models’ (lines 676) when in practice such dichotomy is not established in the previous literature ?.

The investigation should be mainly explanatory, not descriptive. Suggesting a research avenue to develop more ‘descriptive’ views does not seem appropriate (lines 690-691).

What is a ‘critical eye’ (lines 276-277). Please avoid normative and subjective opinions and expressions.

The written English requires major improvements. There are many repetitions. Some expressions are not relevant and/or are not adapted.

Many other parts of the paper require fine tuning. The paper includes major pitfalls. In summary, paper should be entirely rewritten and reconsidered.

The written English requires major improvements. There are many repetitions. Some expressions are not relevant and/or are not adapted.

The style should be improved considerably.

Author Response

Response to Reviewer 1 Comments

Point 1: The title of the paper suggests it is quite ambitious as it intends to provide an examination of the ‘family farmers in short and long food supply chains’.

Response 1: The title of the manuscript has been changed

Point 2: The title and/or the contents of the paper should be aligned. In the body of the text the authors use multiple expressions but the dominant ones are not, by far, the ones quoted in the title of the paper: chain (quoted only 5 times), supply (11 times), channels (quoted 282 times), channel (68 times), marketing channels (2 times), marketing channel (6 times), commercialization (12 times), marketing (52 times), and farmers (122 times). The analysis of the most quoted expression in the body of the text is ‘channels’, not ‘supply chains’. Therefore there seems to be a misfit between the title of the paper and what is developed in the body of the paper. We suggest that the authors define the concepts and harmonize the expressions in the body of the paper and align the title

Response 2: The term supply chain was replaced throughout the text by the term market channel

Point 3: Further, there is a dedicated and mainstream literature on global value chains (GVC) and economic development impacting farmers in developed and developing countries (Gereffi and Korzeniewicz, 1994; Gereffi, 1996; Gereffi et al., 2001; Gereffi, 2005; Humphrey; Gereffi and Lee, 2012, Humphrey and Memedovic, 2006; etc). Why such literature is not taken into consideration in the context of the current paper?

Response 3: Incorporated citation from article "Global value chains and agrifood standards: Challenges and possibilities for smallholders in developing countries" (LEE; GEREFFI; BEAUVAIS, 2012) in the introduction

Point 4: The abstract indicates the study intends to assess the relationships between short and long supply chains to promote or inhibit commercial inclusion. However, in the body of the text we also see a discussion and empirical data related to ‘vulnerability’.

Response 4: The term vulnerability has been deleted from the text. It was replaced by terms that do not carry with them any conceptual load that is outside the scope of the study.

Point 5: The sentence in lines 43 and 44 is not complete (links between agents and ???).

Response 5: The sentence is complete (line 39-41): “The short channels are characterized by direct sales from the producer to the consumer while the long channels are marked by the existence of other links between the production and consumption stages.” 

Point 6: How do you define what an ‘agent’ is ?

Response 6: The conceptualization of agency is not part of the scope of the study. Therefore, we do not consider it necessary to characterize

Point 7: In the general introduction please indicate what refers to ‘self-consumption’ of farmers’ production (see, for example, lines 66-67).

Response 7: This is the text of lines 65 to 67: "The 2017 Brazilian Agricultural Census revealed that 71.83% of family farms in the Brazilian state of Goiás have the market as the main destination of their production, while 28.17% have consumption as the main destination of their production [9]". The term self-consumption is not present in the text

Point 8: The authors justify the interest of the paper (‘research gap’) is based on the lack of ‘precise information in the Agricultural Censuses on how the products produced by family farmers are commercialized, and also, the scientific literature lacks quantitative studies that evaluate the economic relevance between short and long channels’ (lines 74-78). The interests of a paper can be theoretical, practical, and methodological. However, in a science-based paper you are expected to use/introduce one or several research streams to justify your problem statement. This statement related to the unavailability of data can be (partly) accepted for a research paper however you will need to make a literature review of the current literature review and summarize it on a paper. Such statement was not demonstrated in the general introduction or in the section dedicated to the literature review. Please summarize the literature review with previous studies as well as the orientation of the studies (qualitative/quantitative, mixed-methods…) on a table.

Response 8: Changes were made to the text in order to comply with the reviewer's suggestion and new tables and figures were also inserted.

Point 9: At the end of the introduction, you should present thoroughly the organization and the contents of the sections included in the paper

Response 9: We chose not to accept this suggestion in order not to make the text too long. We opted for a more succinct introduction

Point 10: The section dedicated to the literature review includes some general expressions/statements without explaining or defining such concepts. For example, the paragraph in lines 94-100 is merely general/broad.

Response 10: Both this paragraph and the literature review as a whole have been amended.

Point 11: The concept of ‘food orders’ (line 101) is not defined. The sentence referring to the ‘concept of food orders’… ‘takes into consideration the coexistence of several food orders’ (lines 104-106) does not clarifies what a ‘food order’ is.

Response 11: A brief conceptualization of food orders was inserted. In the bibliographical references, the source from which we extracted this concept was changed. We replaced the book written in Portuguese with the book translated into English. Figures were inserted that help to understand aspects involving the concept of food orders

Point 12:The ‘dichotomy’ between short and long chains is repeated multiple times (in the general introduction, in the literature review, etc). (lines 22-23, 151, 199, 248-249…). The word ‘coexistence’ is repeated 56 times (!) in the body of the text. Please avoid repetitions in the body of the text.

Response 12: The number of times the term dichotomy was cited was reduced. The term coexistence is a basic concept of the article, therefore difficult to be replaced by synonyms.

Point 13: Can you summarize on a figure the literature/concepts/comparison related to short and long channels ? (supply chains ?). What are the differences summarizing the different ‘models’ ? (line 115).

Response 13: New figures were inserted to meet this suggestion

Point 14: How do you define what is an ‘inclusive’ and ‘vulnerable’ food system ?. Is a ‘food system’ the same as a ‘food channel’ ? (line 119) and ‘agri-food system’ (line 127) ?, ‘agri-food models’ (line 184)..

Response 14: The term vulnerable has been suppressed in the text. The term inclusive is characterized in the methodology as an agri-food system composed of marketing channels capable of including family farmers.
Two figures were inserted at the beginning of the theoretical framework that illustrate such issues.

Point 15: Please avoid using normative statements in the body of the text. For example, ‘the search for more inclusive and less vulnerable food systems is fundamental…’ (line 119). ‘This inclusive character can be achieved by combining…’ (line 120). These are normative statements, not appropriate for a research paper. Further, such statements were not demonstrated in your literature review or empirical study.

Response 15: Sentences with such normative statements have been deleted from the text.

Point 16: How do you define what is ‘the identity’ of a short channel ? (line 132).

Response 16: The term identity was replaced by the term characteristics

Point 17: After reading the paper, it is difficult to understand what is the conceptual positioning and the theoretical streams developed in the paper. There is no research question and therefore we cannot link it to the methods and to further discussions.

Response 17: In this second round, we seek to qualify the text so that it meets the reviewer's suggestion

Point 18: After reading the paper, it is difficult to understand what is the conceptual positioning and the theoretical streams developed in the paper. There is no research question and therefore we cannot link it to the methods and to further discussions.

Response 18: In this second round, we sought to qualify the text so that it meets the reviewer's suggestion. Especially in the penultimate paragraph of the introduction and the first paragraph of the discussion.

Point 19: The ‘keywords’ presented in the paper are very limited and do not represent the essence of the paper. For example, the two following keywords summarize a similar focus: ‘entrepreneur’ and ‘entrepreneurship’. Other possible keywords could include: ‘Brazil’, ‘survey’, etc.

Response 19: The keyword marketing chanel was removed and the keyword agri-food systems was included. The keyword Brazil was not included, as it was added to the article title.

Point 20: The section dedicated to the literature review should be entirely rewritten. The concepts should be defined; the distinction between short/long chains should be clearly made (build also a figure explaining the dimensions involved, previous studies, etc). The literature review should identify the research gap and to help you to build a conceptual framework to build bridges with the empirical study to be developed in the methods. In the general introduction your research ‘gap’ was mainly defined in relationship with the lack of quantities studies. This argument was not clearly established and discussed in the section dedicated to the literature review. What were the qualitative, quantitative studies and other studies involving mixed-methods that related to the study of short/long supply chains.

Response 20: The difference between short channels and long channels has been reinforced. A table was inserted describing the dimensions that involve the concept of short channels. In the literature review, a text was inserted addressing the gap identified in the literature and the contribution of this article to fill this gap.

Point 21: In the general conclusion, the authors wrote that the study ‘ contributes to the literature by revealing that the consequences of the large participation of family farmers in long channels focused on mono-crop production impacts not only the category of family farmers but also the promotion of resilient agri-food systems’. (lins 664-667). There was not survey of the literature review to discuss the consequences of the participation of family and non-family farmers in the agri-food chains or resilient food systems. In the absence of a systematic review of the literature, the authors cannot advance such conclusions. Further, contrary to a research gap on ‘quantitative studies’ this point was not highlighted in the general introduction as a potential contribution of the paper

Response 21: The term resilience was deleted from the conclusion in the two passages where it was mentioned. The term food safety was maintained.
At the end of the penultimate paragraph, a sentence was inserted citing the intention to contribute to the literature by generating quantitative data in a field of research marked by the predominance of qualitative studies.

Point 22: Why in the conclusion you remind again that your study ‘refutes the dichotomy between models’ (lines 676) when in practice such dichotomy is not established in the previous literature ?.

Response 22: “...refutes the dichotomy between models - long channels are not the antithesis of short channels” This sentence was taken from the conclusion

Point 23: The investigation should be mainly explanatory, not descriptive. Suggesting a research avenue to develop more ‘descriptive’ views does not seem appropriate (lines 690-691).

Response 23: The quoted lines have been deleted from the text.

Point 24: What is a ‘critical eye’ (lines 276-277). Please avoid normative and subjective opinions and expressions.

Response 24: In the first round the term critical eye was replaced by critical view. In this second round we suppress the term critical view.

Reviewer 2 Report

This study provides an interesting topic and appropriate topic.This version of the manuscript has been dramatically improved.I suggest accepting the manuscript.

This study provides an interesting topic and appropriate topic.This version of the manuscript has been dramatically improved.I suggest accepting the manuscript.

Author Response

Response to Reviewer 1 Comments

Point 1: Figure 4 lacks a north arrow.

Response 2: A compass rose has been inserted in the figure

Point 2: The limitations of the methods and models in Discussion.

Response 2: The limitations of the methods and models are present in the conclusion

Round 3

Reviewer 1 Report

‘Family farmers in short and long marketing channels: lessons for rural development in Goias Brazil’.

This paper introduces an important topic for entrepreneurship studies and fits well in the general purpose of the journal. Following the abstract, the objective of the paper is ‘to assess to what extent the relationship between short and long-marketing channels promotes or inhibits the commercial inclusion of family farmers’.

The first round of reviews already produced some significant improvements in the papers. However, despite of the focus of the general purpose of the article, this proposal has many weaknesses: the title is not aligned with the contents, some concepts are not clearly defined, the abstract should be rewritten, the introduction is mainly descriptive, there is limited literature review, the methods should be detailed and improved, external validity was not implemented, the conclusion should be improved…

The abstract should be improved. It is not appropriate to refer to a percentage of results when the sample size was not introduced (lines 10-11).

The end of the general introduction should present the different parts of the paper on an appropriate manner (lines 98-99). The contents of the different parts should be detailed.

The literature review should be developed and completed. It is not appropriate to start the literature review by stating what it is not about (lines 103-105)

The literature review should be focused on the discussions related to inclusion/exclusion of smallholders in global value chains. Further, the literature review should also discuss the ‘measurement’ issues related to inclusion/exclusion. How previous scientific qualitative and quantitative studies measured inclusion/exclusion in global value chains ?.

For example, Ros-Tonen et al.(2019) published a paper documenting ‘Conceptualizing inclusiveness of smallholder value chain integration’. The authors contend in particular that ‘value chain participation and collaboration will never be inclusive at all’ (I), ‘inclusiveness is not a state of being, but mainly a process’ (II); ‘there are multiple pathways towards achieving inclusiveness’ (III).

The current paper neglects in particular that inclusiveness is mainly a ‘process’ and that there are ‘multiple pathways’ towards achieving inclusiveness. In particular, the index defined by the authors in the papers suggests that such measurement is appropriate to understand and capture the essence of inclusiveness. There is no critical view on the establishment of such index. Further, the approach developed by the authors in the paper neglects the ‘process’ approach to achieve inclusiveness. These are two weak points in the general conceptualization of the paper.

The use of capital letters in the references in the body of the text is not appropriate.

The figures 1 and 2 are not developed and discussed in the body of the text. The authors should present a figure that explains how inclusion/exclusion was assessed in different international studies, not some vague informations about the organization of agrifood chains or the agri-food system.

To what extent the ‘food orders’ and ‘food regimes’ are related to the inclusion/exclusion ? how to measure inclusion/exclusion in different food regimes ?

Table 1 is not appropriate. You should focus on measurement issues related to smallholder inclusion/exclusion in value chains, not the dichotomy between the two types of chains.

The literature review should not make reference to Brazil or the state of Goias (lines 239-262). Such information should be introduced in the methods, not the literature review. You should limit your presentation and discussion to the literature review.

What is the scale of the map in figure 3 ?

If you run a cluster analysis please report the percentage of contribution to the axis, the variance, squared-root, the quality of the representation of each variable included in the clster, and the homogenous groups (lines 448-450). This should be reported clearly in the paper.

Figure 5 is difficult to read and to interpret. What the distance means ?

The conclusions should discuss the reliability of the measurements instruments to assess inclusion/exclusion. Further, the ‘process’ approach should also be included.

In addition, nothing is discussed about the revenues captured by smallholder in short supply chains. Is it possible that a smallholder using your approach can be considered ‘excluded’ from the value chains but considering that he/she is selling high-value products, the number of relationships in the chain are not an issue as he/she is still making profit and making his/her living (see, for example, Briones, 2015).

Therefore, the measurement issues are a major issue in this paper and need to be clarified.

The final literature needs to be upgraded. Many relevant references related to global value chains and inclusion/exclusion were not taken into consideration.

The style of English should be improved. The authors use often the past tense. Some expressions should be clarified.

Author Response

1) The abstract should be improved. It is not appropriate to refer to a percentage of results when the sample size was not introduced (lines 10-11)

Response 1: The sample size was inserted in parentheses right after the percentage citation.

2) The end of the general introduction should present the different parts of the paper on an appropriate manner (lines 98-99). The contents of the different parts should be detailed.

Response2: Text was inserted briefly describing each section.

3) The literature review should be developed and completed. It is not appropriate to start the literature review by stating what it is not about (lines 103-105)

Response 3: This sentence was removed from the text. Sentences were inserted in some paragraphs and new paragraphs to make the literature review more complete.

4) The literature review should be focused on the discussions related to inclusion/exclusion of smallholders in global value chains. Further, the literature review should also discuss the ‘measurement’ issues related to inclusion/exclusion. How previous scientific qualitative and quantitative studies measured inclusion/exclusion in global value chains ?

Response 4: Global value chains are outside the scope of our study. Although some long channels are part of global value chains, this is a different topic from the approach we are proposing. We agree that such a discussion would enrich the text, but would make it too long, as we believe that the discussion that should be deepened is that of short food supply chains and the coexistence of different agrifood models. In order to respond to the reviewer's suggestion, we have included a brief discussion on the participation of family farmers in global value chains.

5) For example, Ros-Tonen et al.(2019) published a paper documenting ‘Conceptualizing inclusiveness of smallholder value chain integration’. The authors contend in particular that ‘value chain participation and collaboration will never be inclusive at all’ (I), ‘inclusiveness is not a state of being, but mainly a process’ (II); ‘there are multiple pathways towards achieving inclusiveness’ (III).

Response 5: A paragraph was inserted discussing the process of inclusion/exclusion of family farmers in global value chains.

6) The current paper neglects in particular that inclusiveness is mainly a ‘process’ and that there are ‘multiple pathways’ towards achieving inclusiveness. In particular, the index defined by the authors in the papers suggests that such measurement is appropriate to understand and capture the essence of inclusiveness. There is no critical view on the establishment of such index. Further, the approach developed by the authors in the paper neglects the ‘process’ approach to achieve inclusiveness. These are two weak points in the general conceptualization of the paper.

Response 6: A paragraph was inserted in the methodology emphasizing that the present study did not analyze the process, but a specific moment in time. A paragraph was inserted in the theoretical framework where the idea of inclusion/exclusion as a process was developed.

7) The use of capital letters in the references in the body of the text is not appropriate.

Response 7: Capital letters have been removed from references in the body of the text.

8) The figures 1 and 2 are not developed and discussed in the body of the text. The authors should present a figure that explains how inclusion/exclusion was assessed in different international studies, not some vague informations about the organization of agrifood chains or the agri-food system.

Response 8: At this point in the text, the focus is not on discussing inclusion/exclusion. This discussion is developed later. In this part of the text, the objective is to bring important academic concepts and debates to studies on short channels and their relationships in agrifood systems. The paragraphs about the figures were better developed

9) To what extent the ‘food orders’ and ‘food regimes’ are related to the inclusion/exclusion ? how to measure inclusion/exclusion in different food regimes ?

Response 9: An error occurred in the translation. It was not supposed to be food regimes, but rather food orders (according to the reference NIEDERLE; WESZ JÚNIOR, 2021). All quotes from food regimes have been replaced by food orders. It is not part of the scope of this study to relate the inclusion/exclusion process to the concept of food orders. This concept is cited and briefly characterized, as it helps to understand that short and long channels are not necessarily in diametrically opposed fields and are not necessarily convergent in agrifood systems.

10) Table 1 is not appropriate. You should focus on measurement issues related to smallholder inclusion/exclusion in value chains, not the dichotomy between the two types of chains.

Response 10: There are different conceptualizations of short channels, and we believe it is necessary to make these differences explicit in order to better understand the concept of long channels as well.
The focus of the article is not the participation of family farmers in global value chains, but rather the identification of weaknesses and consistencies in the analyzes that dichotomize or converge the short and long channels within the scope of the participation of family farmers. Our focus is not specifically on value chains.

11) The literature review should not make reference to Brazil or the state of Goias (lines 239-262). Such information should be introduced in the methods, not the literature review. You should limit your presentation and discussion to the literature review.

Response 11: In the methodology, secondary data on the state of Goiás were described. We believe it is necessary to insert the discussion about Goiás in the theoretical framework, as these are relevant theoretical aspects for the study of the coexistence of commercialization channels in the adopted geographical area.

12) What is the scale of the map in figure 3 ?

Response12: In the map figure, the ruler located at the bottom is a graphic scale.

13) If you run a cluster analysis please report the percentage of contribution to the axis, the variance, squared-root, the quality of the representation of each variable included in the clster, and the homogenous groups (lines 448-450). This should be reported clearly in the paper

Response 13: A table was inserted as an appendix with all this information. The text of this paragraph was reworked to meet the reviewer's suggestion.

14) Figure 5 is difficult to read and to interpret. What the distance means ?

Response 14: The figure was redone and a different layout was adopted to improve the interpretation of the dendrogram. The text has been reworked to improve understanding of the meaning of distance.

15) The conclusions should discuss the reliability of the measurements instruments to assess inclusion/exclusion. Further, the ‘process’ approach should also be included.

Response 15: A paragraph (5th paragraph of the conclusion) was inserted to address this consideration.

16) In addition, nothing is discussed about the revenues captured by smallholder in short supply chains. Is it possible that a smallholder using your approach can be considered ‘excluded’ from the value chains but considering that he/she is selling high-value products, the number of relationships in the chain are not an issue as he/she is still making profit and making his/her living (see, for example, Briones, 2015). Therefore, the measurement issues are a major issue in this paper and need to be clarified.

Response 16: A paragraph was inserted at the end of the results presenting data collected by the research on income generation obtained by family farmers in the different channels. A paragraph was added to the methodology explaining the treatment of data on income generation. A paragraph was inserted at the end of the discussion in order to comply with the reviewer's suggestion.

17) The final literature needs to be upgraded. Many relevant references related to global value chains and inclusion/exclusion were not taken into consideration.

Response 17: A few more references have been added. However, we reaffirm that the scope of our study is not global value chains or contract farming, but the relationship between short and long channels. We believe that the theoretical framework adopted combines the recent literature on the coexistence of agrifood models with the seminal literature on short food supply chains.

18) The style of English should be improved. The authors use often the past tense. Some expressions should be clarified.

Response 18: We opted to send the text for review by a professional in the translation of scientific articles after finalizing the adjustments proposed by the reviewers. Therefore, after this review round, we will send the file with the final version of the article to the referred professional to improve the style of English

Round 4

Reviewer 1 Report

‘Family farmers in short and long marketing channels: lessons for rural development in Goias Brazil’.

This paper introduces an important topic for entrepreneurship studies and fits well in the general purpose of the journal. Following the abstract, the objective of the paper is ‘to assess to what extent the relationship between short and long-marketing channels promotes or inhibits the commercial inclusion of family farmers’.

The first two rounds of reviews already produced some significant improvements in the paper. However, despite the improvements, the paper still require significant changes and readjustments.

The literature review should be entirely rewritten. The literature on agri-food models should be discussed and criticized. For example, there was no discussion of the different concepts and perspectives in the agrifood models derived from Niederle and Wesz Junior (2022). Many of the items listed in Figure 2 are not discussed. For example, why do you list Big milk producers and not other types of producers/crops ? Is ‘Big farmers’ the same as ‘Big milk producers’? There is not discussion on the influence of standards, legislation, and inspection and how this issues influence agri-food chains. Many other items would require a deeper discussion.

I did not find the following reference listed in the final references. Niederle and Wesz Junior (2022). These two authors published a paper in 2021. (see line 857)

Lines 128 to 131 should include at least a reference.

The agri-food models included in figure 1 are not discussed in the body of the paper. How this agri-food models compare to other typologies of agri-food models such as the ‘mass market’, ‘Circular Business Model’, the ‘Terroir-based model’ etc (see, for example, Donner and De Vries, 2023).

The figure liste in page 15 is not numbered (‘groupings and sets of marketing channels’).

The reference to the data collection method should be rewritten in line 369.

Why do you use a diferente terminology in the sources ?. For example:

*Survey data collected in 2020 (line 465)

*Data collected by the survey in 2020 (line 494)

Is it the same thing ? The passive voice is not adapted in the second case. The English needs to be reconsidered.

‘Most of the sampled municipalities….’ (line 506). Or, should it be ‘Most of municipalities included in the sample’.

The abbreviations in table 7 should be explained. For example, the following issues should be explained: N, Sum, Mean, Std. error, Stand. Deviation…

Is the world ‘Std’ the same as ‘Stand.’

Why do you use capital letters do describe the variables in table 7. Why in table 6 you use a different format for the variables/channels ?

The description of the information available in table 7 should be improved.

The sentence included in page 571-572 has no verb. (‘are more susceptible to problems’)? Maybe you should use the verb TO HAVE.

Please improve the formalization.

There references at the end of the paper are incomplete, there are some mistakes, and you do not follow entirely the standards of the journal.

There are quite serious English problems in this paper. The paper should be entirely proofread by a native speaker.

There are quite serious English problems in this paper. The paper should be entirely proofread by a native speaker.

Author Response

1) The literature review should be entirely rewritten. The literature on agri-food models should be discussed and criticized. For example, there was no discussion of the different concepts and perspectives in the agrifood models derived from Niederle and Wesz Junior (2022). Many of the items listed in Figure 2 are not discussed. For example, why do you list Big milk producers and not other types of producers/crops ? Is ‘Big farmers’ the same as ‘Big milk producers’? There is not discussion on the influence of standards, legislation, and inspection and how this issues influence agri-food chains. Many other items would require a deeper discussion.

Response 1)

  • Text was inserted (at the end of the first paragraph after figure 2) explaining that the approach to food orders does not prioritize the characterization of production to the detriment of consumption (and vice versa) and that it is not based on the investigation of a production chain specific. A text was inserted (just after figure 1) that brought some more elements of the discussion about agri-food models.
  • I mentioned large milk producers, as this category is common in Goiás. But I agree that it is not relevant to mention it in the study. Big farmers already includes the category that I intended to mention in the figure. I removed the term “Big milk producers” from the figure.
  • The debate about the influence of standards and supervision on the participation of family farmers in markets is quite broad and there are many studies that investigate this. Despite being a topic that concerns our study, it is not part of our scope. A short text mentioning this issue was inserted at the end of the first paragraph after figure 2. Some other items in figure 2 were also briefly discussed.

2) I did not find the following reference listed in the final references. Niederle and Wesz Junior (2022). These two authors published a paper in 2021. (see line 857)

Response 2) It was a typo that has now been corrected. In fact, the publication is from 2021, but it is not an article, but a book.

3) Lines 128 to 131 should include at least a reference.

Response 3) Three new references were inserted in these lines and in the two previous lines.

4) The agri-food models included in figure 1 are not discussed in the body of the paper. How this agri-food models compare to other typologies of agri-food models such as the ‘mass market’, ‘Circular Business Model’, the ‘Terroir-based model’ etc (see, for example, Donner and De Vries, 2023).

Response 4) Agri-food models were better discussed based on Donner's article; De Vries (2023). However, I emphasize that discussions about mass market, circular business model and Terroir based model are not part of our scope.

5) The figure liste in page 15 is not numbered (‘groupings and sets of marketing channels’).

Response 5) Above the figure, on the previous page, is the title of the figure with its number.

6) The reference to the data collection method should be rewritten in line 369.

Response 6) The sentence was reworked and relocated to the end of the previous paragraph.

7) Why do you use a diferente terminology in the sources ?. For example:

*Survey data collected in 2020 (line 465)

*Data collected by the survey in 2020 (line 494)

Is it the same thing ? The passive voice is not adapted in the second case. The English needs to be reconsidered.

Response 7) Both ways of writing refer to the same thing. We standardized the entire text with the format of the first case.

8) Most of the sampled municipalities….’ (line 506). Or, should it be ‘Most of municipalities included in the sample’.

Response 8) We corrected the sentence as suggested by the reviewer

9) The abbreviations in table 7 should be explained. For example, the following issues should be explained: N, Sum, Mean, Std. error, Stand. Deviation…

Is the world ‘Std’ the same as ‘Stand.

Response 9) The variables mentioned were described in the second paragraph after table 5

10) Why do you use capital letters do describe the variables in table 7. Why in table 6 you use a different format for the variables/channels ?

Response 10) We corrected table 7 and rewrote the variable names with capital letters only in the first letters of each variable. An error occurred regarding the format of the variables. At certain moments during the research, we evaluated some channels together and saved the spreadsheets separately. When I requested data on income generation, I made a mistake and selected the wrong spreadsheet. But it has already been corrected. I inserted the Local markets channel individually in table 6.

11) The description of the information available in table 7 should be improved.

The sentence included in page 571-572 has no verb. (‘are more susceptible to problems’)? Maybe you should use the verb TO HAVE.

Response 11) The sentence was corrected and the verb TO HAVE was included.

12) Please improve the formalization.

There references at the end of the paper are incomplete, there are some mistakes, and you do not follow entirely the standards of the journal.

Response 12) A careful review and correction was carried out in accordance with the magazine's standards.

13) There are quite serious English problems in this paper. The paper should be entirely proofread by a native speaker.

Response 13) At the end of all the corrections and adaptations suggested by the reviewer, we sent the text to an English-speaking reviewer who carefully read it. The professional made the necessary corrections to the text, especially with regard to the passive voice (which was cited by the reviewer as a weakness of the paper in the previous round).